

# Characterization of the transcriptional response of Candida parapsilosis to the antifungal peptide MAF-1A

Rong Cheng[1], Wei Li[2], Klarke M. Sample[3,4], Qiang Xu[3,4], Lin Liu[4,5], Fuxun Yu[3,4], Yingjie Nie[3,4], Xiangyan Zhang[4,5] and Zhenhua Luo[3,4]

[1] Guizhou University School of Medicine, Guiyang, China
[2] Department of Cadiovascular Medicine, Affiliated Hospital of Guizhou Medical University, Guiyang, China
[3] Department of Central Lab, Guizhou Provincial People's Hospital, Guiyang, China
[4] NHC Key Laboratory of Pulmonary Immune-related Diseases, Guizhou Provincial People's Hospital, Guiyang, China
[5] Department of Respiratory and Critical Care Medicine, Guizhou Provincial People's Hospital, Guiyang, China

Corresponding authors
Xiangyan Zhang,
zxiangyan12@sina.com
Zhenhua Luo, luo8300@sina.com

## ABSTRACT

Candida parapsilosis is a major fungal pathogen that leads to sepsis. New and more effective antifungal agents are required due to the emergence of resistant fungal strains. MAF-1A is a cationic antifungal peptide isolated from Musca domestica that is effective against a variety of Candida species. However, the mechanism(s) of its antifungal activity remains undefined. Here, we used RNA-seq to identify differentially expressed genes (DEGs) in Candida parapsilosis following MAF-1A exposure. The early (6 h) response included 1,122 upregulated and 1,065 downregulated genes. Late (18 h) responses were associated with the increased expression of 101 genes and the decreased expression of 151 genes. Upon MAF-1A treatment for 18 h, 42 genes were upregulated and 25 genes were downregulated. KEGG enrichment showed that the DEGs in response to MAF-1A were mainly involved in amino acid synthesis and metabolism, oxidative phosphorylation, sterol synthesis, and apoptosis. These results indicate that MAF-1A exerts antifungal activity through interference with Candida parapsilosis cell membrane integrity and organelle function. This provides new insight into the interaction between Candida parapsilosis and this antimicrobial peptide and serves as a reference for future Candida parapsilosis therapies.

# INTRODUCTION

Immunosuppressed patients are at a high risk of hospital-acquired fungal infections. Candida albicans (C. albicans) is the most common pathogen of Candida species, its dominance has decreased as the incidence of non-albicans Candida (NAC) species have increased (*Vieira de Melo et al., 2019*). Over the last two decades, epidemiological studies of fungal pathogens have shown that NAC has surpassed C. albicans as the most prevalent cause of invasive Candida (*Sular et al., 2018*). New anti-NAC treatment regimens are therefore urgently required.

Amongst NAC infections, Candida parapsilosis (C. parapsilosis) is particularly problematic due to its propensity to form biofilms on central venous catheters and other medically implanted devices (*Fais et al., 2017*; *Vieira de Melo et al., 2019*). Additionally, patients in the intensive care unit (ICU) who have undergone total parenteral nutrition are highly susceptible to C. parapsilosis infection, including undernourished children and neonates of low-birth-weights. Recent epidemiological studies have shown that C. parapsilosis is the second most commonly isolated species following only C. albicans in southern Europe, some regions of Asia, and Latin America (*Toth et al., 2019*). When immunosuppressed patients are exposed to C. parapsilosis, the rate of infection is high. The biological characteristics of infection, include toxicity, immune regulation, and drug resistance are in contrast with those of C. albicans (*Toth et al., 2019*). These interspecies specificities affect recognition by the host, clearance, and antifungal drug efficacy.

Candida pathogens have developed varying degrees of drug resistance, with some representing a serious threat to human health (*Robbins, Caplan & Cowen, 2017*). The currently available antifungal agents inhibit cell wall synthesis (echinocandins), destroy cell membrane components (azoles), or bind to ergosterol and perforate the cell membrane (amphotericin B). With the widespread use of antifungal drugs, the presence of drug resistance-related genes has increased. Antimicrobial peptides (AMPs) form a key arm of the innate immune response of a variety of organisms including plants, insects, and humans (*Moravej et al., 2018*). It is uncommon for microbial infections to be resistant to AMPs which are an emerging source of novel antifungal drugs (*Ghosh et al., 2019*; *Nuti et al., 2017*; *Patocka et al., 2018*), making these molecules a potential alternative to fungemia therapies.

AMPs can exhibit both cationic and amphiphilic properties. Cationic AMPs are amphipathic permitting their interaction with negatively charged cell membranes, leading to cell membrane disruption and cell death (*Kobbi et al., 2018*). AMPs are diverse with respect to length (20–100 amino acids), sequence and structure, and are produced by almost all organisms. Filamentous fungi produce a wide spectrum of AMPs that serve as defense and/or host signaling molecules. Penicillium chrysogenum secretes PAF and PAFB that possess complex tertiary structures and activity centers. PAF and PAFB are produced as 92 amino acid preproteins that are active against a variety of pathogenic fungi, bacteria, and viruses (*Huber et al., 2020*). Insects are extremely resistant to microbial infections, which are an important source of AMPs. Insect AMPs are smaller (between 12 and 50 amino acids) with secondary structures formed predominantly of $\alpha$-helices and $\beta$-sheets. Whilst membrane damage is the canonical mechanism through which AMPs act, other mechanisms exist. AMPs have specific subcellular targets, including the inhibition of DNA synthesis, RNA synthesis, protein synthesis, and cell wall integrity (*Guilhelmelli et al., 2013*; *Li et al., 2016*). However, their mechanism(s) of action at the molecular level remain unclear. The Musca domestica antifungal peptide-1 (MAF-1) is a novel cationic AMP isolated from the instar larvae of houseflies (*Fu, Wu & Guo, 2009*). We previously cloned the full-length MAF-1 gene and derived 26 amino acid MAF-1A peptides from the MAF-1 structural domain. Despite the established antifungal effects of MAF-1A, the molecular mechanism(s) governing its activity remain largely undefined (*Zhou et al., 2016*).

In recent years, the development of high throughput sequencing technologies has facilitated research on both antimicrobial drug function and drug-resistance. For example, HAC1 (CPAR2_103720) is a key mediator of endoplasmic reticulum stress in C. parapsilosis identified through RNA-seq (*Iracane et al., 2018*). In our previous studies, we showed that MAF-1A inhibits C. albicans through its effects on the cell wall, plasma membrane, protein synthesis, and energy metabolism (*Wang et al., 2017*). However, the mechanism(s) through which C. albicans responds to MAF-1A were not fully defined. Here, we have expanded our knowledge on how MAF-1A acts on C. parapsilosis and investigated differences in the responses of C. albicans and C. parapsilosis to MAF-1A treatment. RNA-seq was used to investigate changes in gene expression at early (6 h) and late (18 h) time points, according to time-kill curves of C. parapsilosis growth.

## MATERIALS AND METHODS

### Strains and growth conditions

Transcriptional profiling was performed on the C. parapsilosis reference strain ATCC22019. The strain was preserved in goat blood and stored at $-80\,°C$. C. parapsilosis was streaked on Sabouraud Dextrose Agar (SDA) plates (Sangon, Shanghai, China) at $35\,°C$ as described by *Lis et al. (2010)*. MAF-1A treatments were performed in Sabouraud Dextrose Broth (SDB) (Sangon, Shanghai, China).

### Peptide synthesis

MAF-1A was synthesized by Sangon Biotech (Shanghai, Shanghai, China) as a linear peptide of 26 amino acids: KKFKETADKLIESAKQQLESLAKEMK. Analytical high-performance liquid chromatography (HPLC) was used to confirm purity $\geq 95\%$. The peptide was dissolved in sterile ultrapure water at 5 mg/mL and stored at $-20\,°C$.

### Minimum inhibitory concentration (MIC) and time-kill curves

Antifungal assays were performed as per the requirements of the Clinical and Laboratory Standards Institute (CLSI) M27-A3. Briefly, cultures were grown for 24 h at $35\,°C$ and resuspended in SDB. Concentrations were adjusted to approximately $0.5 \times 10^3$–$2.5 \times 10^3$ CFU/mL and 100 μl of the suspension was added to each well of 96-well polypropylene microplates (NEST, Wuxi, China). MAF-1A was added at concentration ranging from 0.1 mg/mL to 1.2 mg/mL. All experiments were performed in triplicates. After incubation at $35\,°C$ for 24 h, absorbances were measured at 492 nm on a Microplate Reader (BioTek Synergy H1, Vermont, USA). MIC was defined as the lowest drug concentration showing 80% growth inhibition compared to the drug-free controls. The following formulas were used (*Li et al., 2008*):

(1) Percentage Fungal Growth = (Treatment Well A Value − Control Well A Value)/(Growth in Control Well A Value − Control Well A Value) × 100%.

(2) Percentage Inhibition of Fungal Growth = 1 − Percentage Fungal Growth.

Time-kill curves were performed according to the literature (*Li et al., 2008*; *Sun et al., 2008*). C. parapsilosis suspensions were mixed with MAF-1A (MIC) in triplicate and cultured at $35\,°C$. Aliquots of 100 μl were removed from each test solution at pre-determined time points (0, 2, 4, 6, 8, 10, 12, 14, 16, 18, 20, 22, and 24 h). Dilutions were

produced (1:100) and streaked in triplicate onto SDA agar plates for colony counts after incubation at 35 °C for 24 h. Sterile ultrapure water was used as a control.

## Transcriptome sequencing

C. parapsilosis was inoculated into SDB medium (Sangon, Shanghai, China) at 35 °C for 24 h. C. parapsilosis was treated with MAF-1A at MIC for 6 h (CPAS) and 18 h (CPBS), before RNA extraction. Untreated cultures served as controls (6 h, CPAC; 18 h, CPBC). Total RNA was extracted using RNAiso Plus (Takara, Dalian, China) according to the manufacturer's instructions. RNA concentration and quality were determined on a NanoDrop 2000 (Thermo Fisher Scientific, Wilmington, DE, USA) and Agilent 2100 bioanalyzer (Agilent Technologies, CA, USA). Libraries were prepared using NEBNext® UltraTM RNA Library Prep Kit (NEB, USA) as per the manufacturer's recommendations. Purified libraries were quantified on an Agilent 2100 bioanalyzer Effective concentrations were determined through qRT-PCR analysis. Libraries were prepared and sequenced using a Novoseq sequencer (Illumina, USA) to produce 150 bp paired-end reads.

## Differential expression analysis

Raw reads were filtered to obtain high-quality clean reads for subsequent analysis. All reads were mapped to the reference genome of C. parapsilosis (assembly ASM18276v2) from the National Center for Biotechnology Information (NCBI) using HISAT2 v2.0.5 (*Kim, Langmead & Salzberg, 2015*). Differential expression analysis between the conditions was assessed using the Bioconductor software package DESeq2 in R 1.16.1 (*Love, Huber & Anders, 2014*). Relative gene expression was assessed using FPKM (Fragments Per Kilobase of transcript sequence per Millions of base pairs sequenced) and compared using log2 FC. *P*-values were adjusted to generate false discovery rates (padj) as described by Benjamini-Yekutieli, assigning the significance threshold for DEGs as padj < 0.05 (*Benjamini et al., 2001*; *Mortazavi et al., 2008*).

## Enrichment and interaction network analysis of the differentially expressed genes

To further understand the functions of the DEGs, Gene Ontology (GO) enrichment was performed using the Bioconductor software clusterProfiler 3.4.4 in R package (*Yu et al., 2012*). Statistical enrichment of the DEGs was also performed in the Kyoto Encyclopedia of Genes and Genomes (KEGG) for pathway enrichment (*Kanehisa et al., 2019*; *Ogata et al., 1999*). PPI analysis of the DEGs was performed based on the STRING database to define key protein-protein interactions (*Yu et al., 2012*). The network was constructed using Cytoscape 3.6.1 (*Shannon et al., 2003*).

## Validation of RNA-seq by quantitative RT-PCR (qRT-PCR)

To confirm the RNA-seq data, 20 DEGs (10 with increased expression and 10 with decreased expression) were selected for qRT-PCR validations. Reactions were performed using SYBR Premix Ex Taq TM Kit (Takara) according to the manufacturer's protocol. Reaction conditions were as follows: 40 cycles of 95 °C for 30 s; 95 °C for 5 s; and 60 °C for 30 s. PCRs were performed on a BIO-RAD CFX-Connect Real-Time System. Relative gene
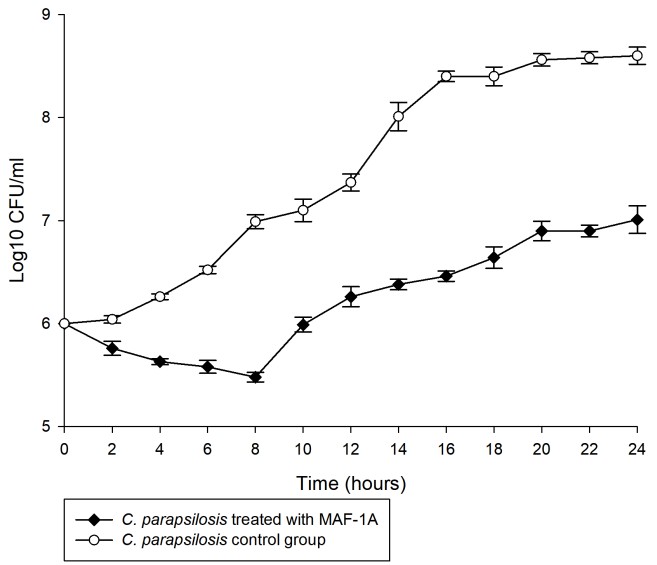

**Figure 1** **Time-kill curves of MAF-1A under MIC for C. parapsilosis.** The mean growth of three C. parapsilosis cultures were recorded (Log10 CFU/ml) every 2 h for 24 h.

expression was determined using the $2^{-\Delta\Delta Ct}$ method normalized to 18S rRNA (*Livak & Schmittgen, 2001*). Significant differences were determined using a $t$-test with a threshold of $p < 0.05$. Primers are listed in Table S1. Primer efficiency and melting curves are listed in Table S2 and Figs. S1–S5.

# RESULTS

## MIC assays and time-kill curves

The MIC of MAF-1A against C. parapsilosis was determined as 0.6 mg/mL. Time-kill curves of MAF-1A at MIC showed a gradual antifungal effect during the first 8 h of C. parapsilosis culture (Fig. 1). After 8 h, cell numbers increased but remained lower than those of the control group.

## Transcriptional stress responses and enrichment analysis of MAF-1A treated C. parapsilosis

RNA-seq analysis in C. parapsilosis treated with MAF-1A for 6 and 18 h showed 5,747 DEGs. Sequence reads were deposited in the NCBI Sequence Read Archive (SRA) under accession number PRJNA638006. A total of 2,439 DEGs were detected. Out of these genes, 2187 were identified at 6 h, representing 38.05% of the total detectable genes. A total of 252 genes were differentially expressed after 18 h and accounted for 4.38% of the total expressed genes. After 6 and 18 h of MAF-1A treatment, 67 DEGs with opposite trends were observed (reversed genes 1, RG1 and reversed genes 2, RG2). In total, 56 DEGs were upregulated, whilst down-regulated genes remained unchanged (one unchanged genes: UG1; two unchanged genes: UG2) (Fig. 2).
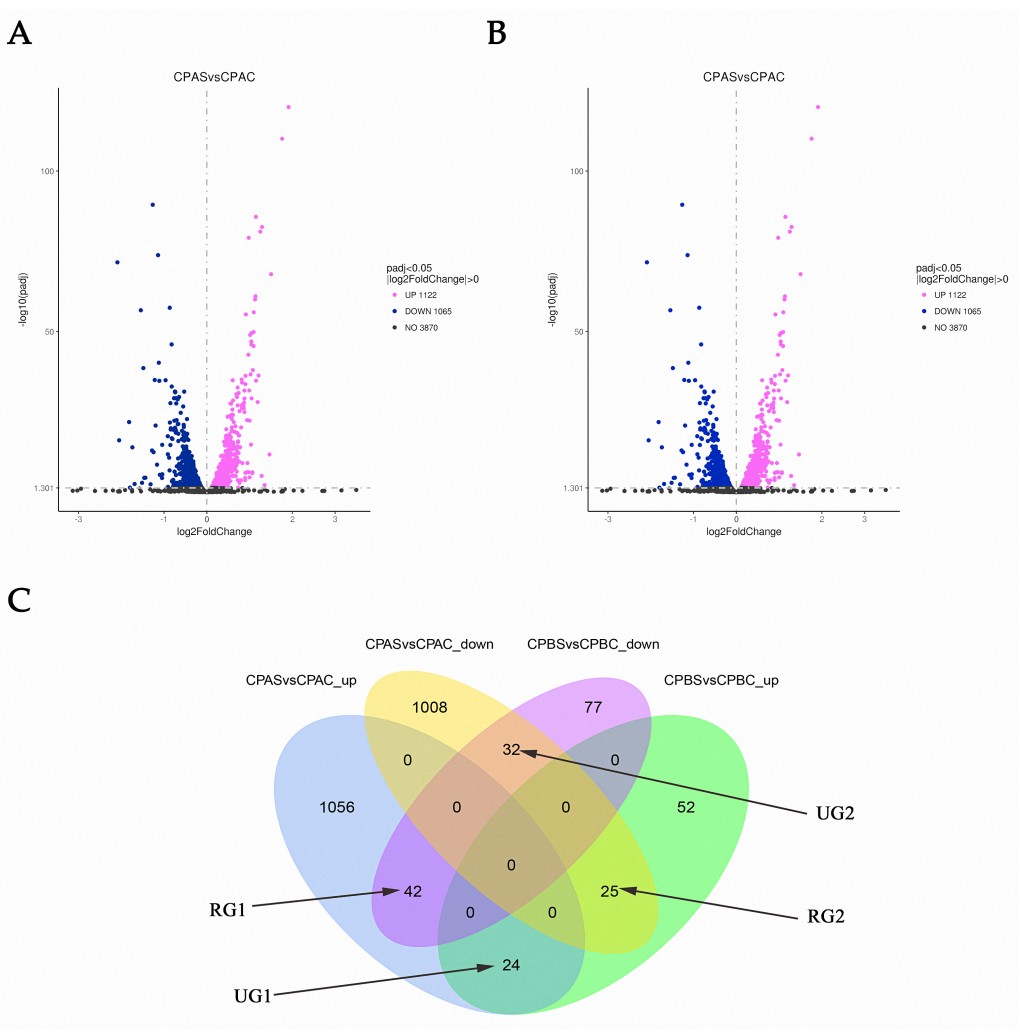

**Figure 2** **Gene expression changes in C. parapsilosis following MAF-1A treatment.** Volcano plots of the DEGs. (A) Volcano plots depicting log2 FC (fold change) in expression after 6 h of treatment with MAF-1A, (C) parapsilosis was treated with MAF-1A at MIC for 6 h (CPAS), without MAF-1A as a control (CPAC). The expression of 1122 genes significantly increased; 1065 genes were significantly downregulated (padj < 0.05). (B) Volcano plot depicting the log2 FC expression after 18 h of treatment with MAF-1A. (C) parapsilosis was treated with MAF-1A at MIC for 18 h (CPBS). Controls lacked MAF-1A treatment (CPBC). The expression of 101 genes significantly increased in contrast to 151 genes whose expression decreased (padj < 0.05). (C) Gene expression Venn diagrams revealing two gene groups with opposite trends, labeled as RG1, RG2, UG1 and UG2; CPAS vs. CPAC_up: genes with increased expression after 6 h; CPAS vs. CPAC down: genes with decreased expression after 6 h; CPBS vs. CPBC_up: genes with increased expression after 18 h; CPBS vs. CPBC down: genes with decreased expression after 18 h.

## *DEG enrichment analysis*

Amongst the DEGs at 6 h, 1122 showing increased expression were enriched in 85 KEGG pathways, 20 of which were significant with padj < 0.05. The most significant pathways with increased expression in C. parapsilosis following MAF-1A treatment were: oxidative

**Table 1 Significantly enriched KEGG pathways for genes with increased expression after 6 h of MAF-1A treatment.**

| KEGG ID | Description | $p$ value | padj |
|---|---|---|---|
| cdu00190 | Oxidative phosphorylation | $3.34 \times 10^{-12}$ | $2.84 \times 10^{-10}$ |
| cdu04146 | Peroxisome | $2.02 \times 10^{-8}$ | $8.56 \times 10^{-7}$ |
| cdu00020 | Citrate cycle (TCA cycle) | $2.21 \times 10^{-5}$ | $6.27 \times 10^{-4}$ |
| cdu01200 | Carbon metabolism | $3.50 \times 10^{-5}$ | $7.44 \times 10^{-4}$ |
| cdu04111 | Cell cycle—yeast | $9.20 \times 10^{-5}$ | $1.56 \times 10^{-3}$ |
| cdu04011 | MAPK signaling pathway—yeast | $8.17 \times 10^{-4}$ | $1.16 \times 10^{-2}$ |
| cdu04113 | Meiosis—yeast | $2.35 \times 10^{-3}$ | $2.80 \times 10^{-2}$ |
| cdu00071 | Fatty acid degradation | $2.63 \times 10^{-3}$ | $2.80 \times 10^{-2}$ |
| cdu04136 | Autophagy—other | $4.03 \times 10^{-3}$ | $3.80 \times 10^{-2}$ |

**Notes.**
padj of $< 0.05$ set as the significance threshold.

phosphorylation, peroxisome, citrate cycle (TCA cycle), carbon metabolism, cell cycle-yeast, MAPK signaling-yeast, meiosis-yeast, fatty acid degradation, and autophagy (Table 1). Genes of decreased expression were enriched in steroid biosynthesis, biosynthesis of amino acids, cysteine and methionine metabolism, biosynthesis of antibiotics, ribosome, RNA polymerase, biosynthesis of secondary metabolites, RNA transport, ribosome biogenesis in eukaryotes, lysine biosynthesis, 2-oxocarboxylic acid metabolism, pyrimidine metabolism, glycine, serine and threonine metabolism and purine metabolism (Table 2). At 18 h, 101 genes were upregulated and enriched in 24 KEGG pathways, two of which were significant (Table 3). A total of 151 genes were downregulated and significantly enriched in carbon metabolism, biosynthesis of antibiotics, oxidative phosphorylation, the biosynthesis of secondary metabolites, and the biosynthesis of amino acids (Table 4).

### RG1, RG2, UG1, and UG2 gene enrichment analysis

The 42 genes in RG1 were enriched in 17 KEGG pathways, of which oxidative phosphorylation was most significant. Additionally, 25 genes in RG2 were enriched in 13 KEGG pathways, of which arginine biosynthesis, the biosynthesis of antibiotics, the biosynthesis of amino acids, and the biosynthesis of secondary metabolites were enriched. A total of 24 genes in UG1 and 32 genes in UG2 were enriched in 8 and 20 KEGG pathways, respectively. In UG1, the genes were enriched in butanoate metabolism, propionate metabolism, beta-alanine metabolism, valine, leucine, and isoleucine degradation. No pathways were significantly enriched in UG2 at a padj $< 0.05$ (Table 5).

Genes in RG1, RG2, UG1, and UG2 were enriched in 535 GO terms, a total of 9 of which were significant (Table 6). Genes in RG1 were involved in energy production and redox processes. Genes in RG2 were associated with the anabolic processes of various organic acids. Genes in UG2 were involved in oxidation–reduction processes.

### Verification of differentially expressed genes

A total of 20 genes were selected, including 10 with increased expression and 10 with decreased expression. Genes were evenly selected from 6 h and 18 h time points to validate the RNA-seq data by qRT-PCR. The results indicated the expression levels have a

**Table 2** Significantly enriched KEGG pathways for genes with decreased expression after 6 h of MAF-1A treatment.

| KEGG ID | Description | $p$ value | padj |
|---|---|---|---|
| cdu00100 | Steroid biosynthesis | $1.64 \times 10^{-8}$ | $1.40 \times 10^{-6}$ |
| cdu01230 | Biosynthesis of amino acids | $5.95 \times 10^{-7}$ | $2.53 \times 10^{-5}$ |
| cdu00270 | Cysteine and methionine metabolism | $4.80 \times 10^{-6}$ | $1.36 \times 10^{-4}$ |
| cdu01130 | Biosynthesis of antibiotics | $1.07 \times 10^{-5}$ | $2.27 \times 10^{-4}$ |
| cdu03010 | Ribosome | $3.01 \times 10^{-5}$ | $5.12 \times 10^{-4}$ |
| cdu03020 | RNA polymerase | $2.14 \times 10^{-4}$ | $3.03 \times 10^{-3}$ |
| cdu01110 | Biosynthesis of secondary metabolites | $3.68 \times 10^{-4}$ | $4.47 \times 10^{-3}$ |
| cdu03013 | RNA transport | $9.34 \times 10^{-4}$ | $8.83 \times 10^{-3}$ |
| cdu03008 | Ribosome biogenesis in eukaryotes | $9.35 \times 10^{-4}$ | $8.83 \times 10^{-3}$ |
| cdu00300 | Lysine biosynthesis | $2.31 \times 10^{-3}$ | $1.96 \times 10^{-2}$ |
| cdu01210 | 2-Oxocarboxylic acid metabolism | $2.66 \times 10^{-3}$ | $2.05 \times 10^{-2}$ |
| cdu00240 | Pyrimidine metabolism | $3.54 \times 10^{-3}$ | $2.40 \times 10^{-2}$ |
| cdu00260 | Glycine, serine and threonine metabolism | $3.67 \times 10^{-3}$ | $2.40 \times 10^{-2}$ |
| cdu00230 | Purine metabolism | $5.08 \times 10^{-3}$ | $3.09 \times 10^{-2}$ |

**Notes.**
padj of $< 0.05$ set as the significance threshold.

**Table 3** Significantly enriched KEGG pathways for genes with increased expression after 18 h of MAF-1A treatment.

| KEGG ID | Description | $p$-value | padj |
|---|---|---|---|
| cdu00220 | Arginine biosynthesis | $3.01 \times 10^{-4}$ | $7.21 \times 10^{-3}$ |
| cdu00250 | Alanine, aspartate and glutamate metabolism | $1.48 \times 10^{-3}$ | $1.77 \times 10^{-2}$ |

**Notes.**
padj of $< 0.05$ were set as the significance threshold.

consistent change for both RNASeq and qRT-PCR. Hence, the qRT-PCR results confirmed the reliability of our RNA-Seq data (Fig. 3).

## Protein-protein interaction (PPI) network analysis

We constructed a PPI network based on the STRING database of the DEGs after 6 h of treatment. The PPI network contained 624 nodes and 6264 edges, with a degree filter of $\geq 10$ (Fig. 4). The connectivity degree (dg) of multiple nodes in the PPI network were high, including: CpUbi1 (dg $= 146$), CpGlt1 (dg $= 82$), CpCdc28 (dg $= 54$), CpCys4 (dg $= 50$), CpCyt1 (dg $= 45$), CpRpc40 (dg $= 42$), CpArx1 (dg $= 42$), CpDim1 (dg $= 41$), CpYtm1 (dg $= 41$), CpRip1 (dg $= 41$). Upon enrichment analyses the identified genes were associated with oxidative phosphorylation (Fig. S6).

## DISCUSSION

C. parapsilosis is one of the most prevalent fungal species in many regions. In addition to its high rates of infection, its etiology differs from that of C. albicans (*Holland et al., 2014*). Specific C. parapsilosis isolates are resistant to conventional antifungal drugs including echinocandins, azoles, and amphotericin B (*Lotfali et al., 2016*; *Maria et al., 2018*; *Thomaz et al., 2018*). Antimicrobial peptides lead to cell lysis and death through

**Table 4 Significantly enriched KEGG pathways for genes with decreased expression after 18 h of MAF-1A treatment.**

| KEGG ID | Description | $p$-value | padj |
|---------|-------------|-----------|------|
| cdu01200 | Carbon metabolism | $1.65 \times 10^{-8}$ | $7.61 \times 10^{-7}$ |
| cdu01130 | Biosynthesis of antibiotics | $6.21 \times 10^{-7}$ | $1.43 \times 10^{-5}$ |
| cdu00190 | Oxidative phosphorylation | $5.58 \times 10^{-6}$ | $6.91 \times 10^{-5}$ |
| cdu01110 | Biosynthesis of secondary metabolites | $6.01 \times 10^{-6}$ | $6.91 \times 10^{-5}$ |
| cdu00010 | Glycolysis/Gluconeogenesis | $1.71 \times 10^{-5}$ | $1.57 \times 10^{-4}$ |
| cdu00260 | Glycine, serine and threonine metabolism | $5.83 \times 10^{-4}$ | $4.47 \times 10^{-3}$ |
| cdu01230 | Biosynthesis of amino acids | $1.77 \times 10^{-3}$ | $1.16 \times 10^{-2}$ |
| cdu00680 | Methane metabolism | $3.62 \times 10^{-3}$ | $2.05 \times 10^{-2}$ |
| cdu00520 | Amino sugar and nucleotide sugar metabolism | $4.02 \times 10^{-3}$ | $2.05 \times 10^{-2}$ |
| cdu00052 | Galactose metabolism | $6.24 \times 10^{-3}$ | $2.87 \times 10^{-2}$ |
| cdu00730 | Thiamine metabolism | $7.93 \times 10^{-3}$ | $3.20 \times 10^{-2}$ |
| cdu00630 | Glyoxylate and dicarboxylate metabolism | $8.34 \times 10^{-3}$ | $3.20 \times 10^{-2}$ |
| cdu00330 | Arginine and proline metabolism | $9.61 \times 10^{-3}$ | $3.24 \times 10^{-2}$ |
| cdu00670 | One carbon pool by folate | $9.85 \times 10^{-3}$ | $3.24 \times 10^{-2}$ |

Notes.
  padj of < 0.05 set as the significance threshold.

**Table 5 Significantly enriched KEGG pathways for genes in RG1, RG2 and UG1.**

| Sort | KEGG ID | Description | $p$-value | padj |
|------|---------|-------------|-----------|------|
| RG1 | cdu00190 | Oxidative phosphorylation | $1.34 \times 10^{-5}$ | $2.27 \times 10^{-4}$ |
| RG2 | cdu00220 | Arginine biosynthesis | $1.97 \times 10^{-5}$ | $2.56 \times 10^{-4}$ |
| | cdu01130 | Biosynthesis of antibiotics | $4.09 \times 10^{-3}$ | $2.39 \times 10^{-2}$ |
| | cdu01230 | Biosynthesis of amino acids | $5.52 \times 10^{-3}$ | $2.39 \times 10^{-2}$ |
| | cdu01110 | Biosynthesis of secondary metabolites | $1.21 \times 10^{-2}$ | $3.92 \times 10^{-2}$ |
| UG1 | cdu00650 | Butanoate metabolism | $1.82 \times 10^{-2}$ | $4.91 \times 10^{-2}$ |
| | cdu00640 | Propanoate metabolism | $2.48 \times 10^{-2}$ | $4.91 \times 10^{-2}$ |
| | cdu00410 | beta-Alanine metabolism | $2.64 \times 10^{-2}$ | $4.91 \times 10^{-2}$ |
| | cdu00280 | Valine, leucine and isoleucine degradation | $2.81 \times 10^{-2}$ | $4.91 \times 10^{-2}$ |

Notes.
  padj of < 0.05 set as the significance threshold.

cell membrane leakage (*Papo & Shai, 2003*; *Paterson et al., 2017*; *Shai, 1999*; *Utesch et al., 2018*). However, mechanistic studies of antimicrobial peptides have determined that their membrane interactions are complex. *Park, Kim & Kim (1998)* showed that buforin II prevents microorganisms entry into cells. *Lee et al. (2019)* found that antifungal $\beta$-peptides cause cell death by entering cells and causing nuclear and vacuole dysfunction. *Chileveru et al. (2015)* showed that human alpha-defensin 5 enters the cytoplasm of Escherichia coli and interferes with cell division.

AMPs work through various mechanisms. In our previous studies, we showed that MAF-1A inhibits C. albicans through its effects on the cell wall, cell membrane, and ribosomes (*Wang et al., 2017*). In this study, we found that MAF-1A alters gene expression in several important biological pathways in C. parapsilosis, including oxidation–reduction
**Table 6  Significant enriched GO terms of RG1, RG2 and UG2.**

| Sort | Category | GO ID | Description | *p*-value | padj |
|------|----------|-------|-------------|-----------|------|
| RG1 | BP | GO:0006091 | Generation of precursor metabolites and energy | $8.71 \times 10^{-5}$ | $7.14 \times 10^{-3}$ |
| | BP | GO:0055114 | Oxidation–reduction process | $8.49 \times 10^{-4}$ | $3.48 \times 10^{-2}$ |
| RG2 | BP | GO:0016053 | Organic acid biosynthetic process | $5.27 \times 10^{-4}$ | $2.11 \times 10^{-2}$ |
| | BP | GO:0046394 | Carboxylic acid biosynthetic process | $5.27 \times 10^{-4}$ | $2.11 \times 10^{-2}$ |
| | BP | GO:0044283 | Small molecule biosynthetic process | $1.56 \times 10^{-3}$ | $3.12 \times 10^{-2}$ |
| | BP | GO:0006082 | Organic acid metabolic process | $2.34 \times 10^{-3}$ | $3.12 \times 10^{-2}$ |
| | BP | GO:0019752 | Carboxylic acid metabolic process | $2.34 \times 10^{-3}$ | $3.12 \times 10^{-2}$ |
| | BP | GO:0043436 | Oxoacid metabolic process | $2.34 \times 10^{-3}$ | $3.12 \times 10^{-2}$ |
| UG2 | BP | GO:0055114 | Oxidation–reduction process | $3.53 \times 10^{-4}$ | $2.15 \times 10^{-2}$ |

Notes.
padj of < 0.05 set as the significance threshold for enrichment.

processes and alternative energy sources. We further compared the response of C. albicans and C. parapsilosis to MAF-1A, most DEGs have the same expression trend (upregulated/downregulated), and identified changes in both stress and energy metabolism pathways (carbon metabolism, cell cycle, peroxisome, carbon metabolism, fatty acid degradation) (Tables S3 and S4). We hypothesized that the antifungal peptide MAF-1A exerts antifungal effect and disrupts energy metabolism by affecting oxidation–reduction processes, due to its effects on the mitochondria. Whilst antimicrobial peptides have multiple modes of action, these remain undetermined for MAF-1A. Our findings suggest that intracellular targets may be the key sites of MAF-1A activity, with enrichment analysis of the DEGs suggesting that MAF-1A exerts antimicrobial activity through a variety of mechanisms.

**Membrane destruction**

Genes with decreased expression after 6 h were significantly enriched in steroid biosynthesis (Fig. 5) including *CpERG1*, *CpERG3*, *CpERG6*, *CpERG7*, *CpERG9*, *CpERG11*, *CpERG25*, *CpERG26*, *CpERG27*, *CpERG2*, *CpERG4*, *CpERG5*, *CpERG24*, and *CpSPBC16A3.12c* (*Kanehisa et al., 2019*; *Ogata et al., 1999*). Azole agents exert antifungal activity by inhibiting the synthesis of ergosterol, a major component of fungal cell membranes (*Ermakova & Zuev, 2017*). The overexpression of *ERG11* (encoding lanosterol 14-demethylase) is a major cause of azole resistance, often mediated by point mutations in the ERG11 gene. Members of the *ERG* gene family encode proteins involved in ergosterol biosynthesis, of which lanosterol 14-demethylase is critical. In this study, MAF-1A decreased the expression of 14 genes related to sterol synthesis including *CpERG11*, suggesting it interferes with ergosterol synthesis. Additionally, 10 genes showed increased expression after 6 h and were enriched in fatty acid degradation pathways (Fig. 6) (*Kanehisa et al., 2019*; *Ogata et al., 1999*). Of these, the expression of *CpPOX4* (CPAR2-807700) significantly increased, (log2 FC = 1.503). We further verified the upregulation of these genes through qRT-PCR (log2FC = 2.834). *CpPOX4* encodes a component of fatty acid biosynthesis, which indicates that the composition of the cell membrane was affected by MAF-1A.
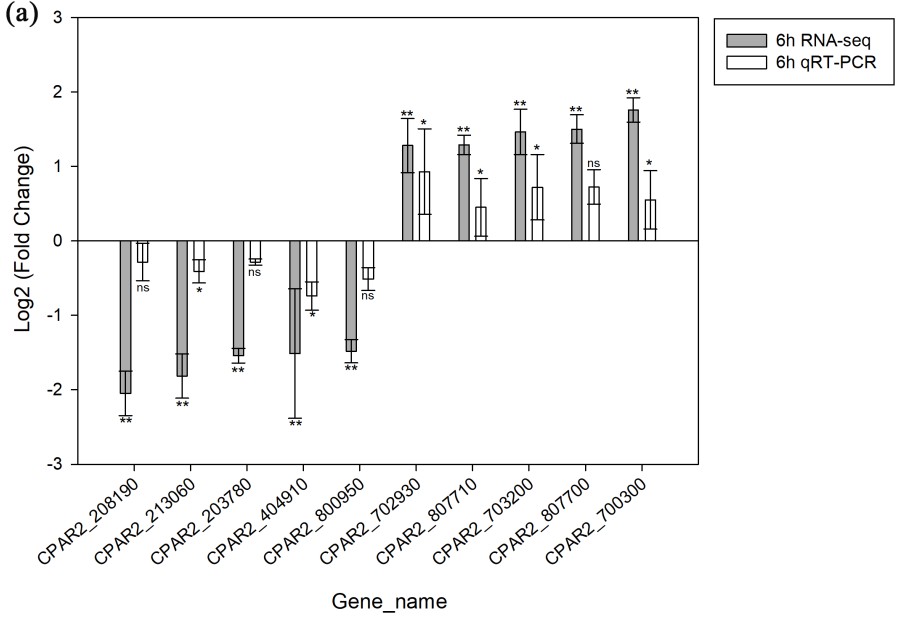

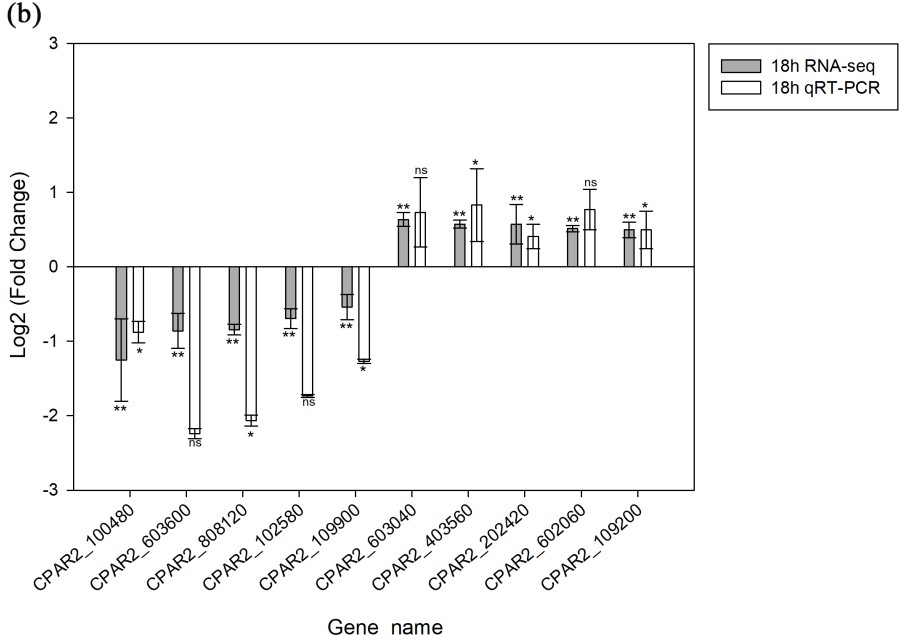

**Figure 3** **Validation of the RNA-seq data via qRT-PCR analysis.** (A) 6 h timepoint. (B) 18 hour time-point. $^*$ $p < 0.05$, $^{**}$ $p < 0.01$, ns, not-significant.

## Mitochondrial function

A total of 42 genes showed increased expression at 6 h and were enriched in oxidative phosphorylation pathways. The genes were mainly involved in respiratory chain electron transport processes in the mitochondrial inner membrane, including NHD release of H+ and ATP synthase (Fig. 7) (*Kanehisa et al., 2019*; *Ogata et al., 1999*). A total of six genes

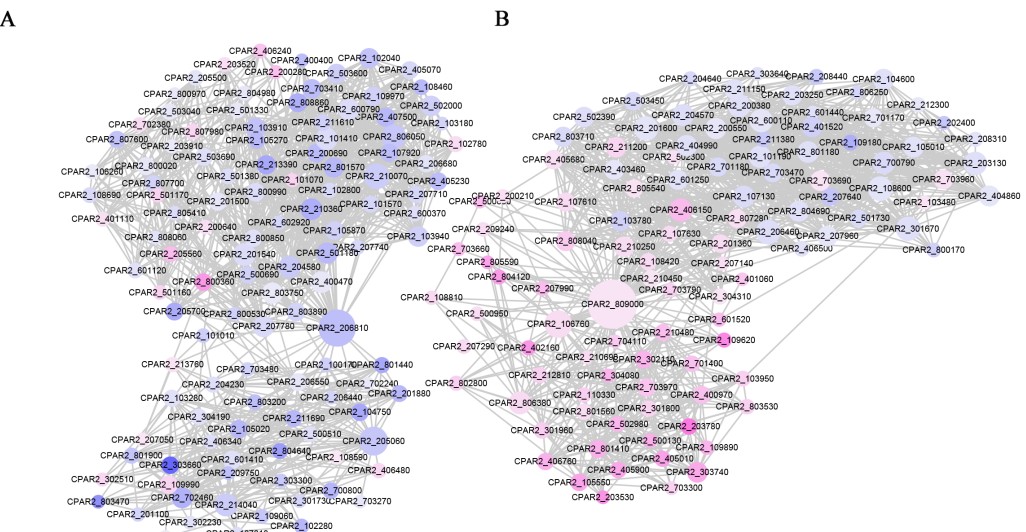

**Figure 4** **PPI network of the DEGs following MAF-1A treatment of C. parapsilosis for 6 h.** Node sizes correlate with node importance; purple nodes denote genes with increased expression, and blue denote decreased expression genes.

in RG1 were also enriched in this pathway (*CpCOX15*, *Cpnuo-21*, *CpQCR2*, *CpQCR8*, *CpQCR7*, and *CpCOR1*). Previous studies showed that *COX15* encodes an indispensable mitochondrial protein for Saccharomyces cerevisiae cytochrome oxidase (*Glerum et al., 1997*). Cytochrome oxidase is a terminal enzyme in the respiratory electron transport chain that is essential for ATP synthesis. Reactive oxygen species (ROS) are produced by the oxidative phosphorylation of ATP and can disrupt the electron transport chain in mitochondria (*Piippo et al., 2018*). ROS production induces damage to lipids, proteins, lipids, and nucleic acids, leading to cell death. Eukaryotes prevent cell damage through oxidative stress detoxification and the prevention of ROS accumulation. In this study, GO enrichment analysis of the RG1 genes showed that 7 that were highly expressed were associated with redox processes (*CpPOX9*, *CpCOX15*, *Cpnuo-21*, *CpAAEL001134*, *CpQCR7*, *CpRGI1*, and *CpnamA*). These processes help cells to remove accumulated ROS. Due to the increased expression of these genes in response to MAF-1A, it is possible that MAF-1A promotes oxidative phosphorylation, which disrupts electron transfer in the mitochondria, enhancing ROS production and subsequent cell damage.

C.parapsilosis has an unusual mitochondrial genome architecture, consisting of linear DNA molecules of 30.9-Kbp, terminating with specific telomeric structures on both sides (738-Kbp long). This differs from telomeres at the ends of eukaryotic nuclear chromosomes, particularly in humans (*Kovac, Lazowska & Slonimski, 1984*). MAF-1A interferes with the expression of multiple genes related to the mitochondrial functions of C. parapsilosis, It is, therefore, feasible that MAF-1A interferes with the normal function of C. parapsilosis mitochondria.

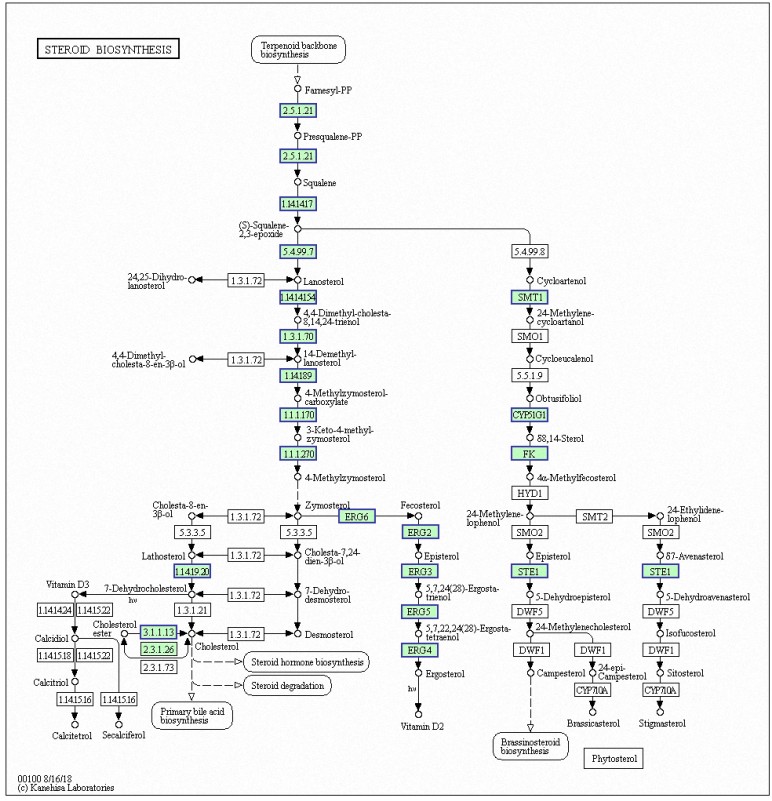

**Figure 5** **Significantly enriched KEGG pathways in steroid biosynthesis.** DEGs with decreased expression are shown in blue. Permission for publication granted by the Kyoto Encyclopedia of Genes and Genomes, Kyoto University, Japan.

## CONCLUSIONS

In summary, MAF-1A has a complex response in C. parapsilosis. Most DEGs identified through RNA-seq analysis were related to oxidation–reduction processes and the use of alternative energy sources, Mitochondria are important target for the anti-fungal peptide MAF-1A to exert anti-C. parapsilosis. RNA-seq data therefore provide future direction to study the antifungal mechanisms of MAF-1A and highlight the potential pathways that contribute to resistance.

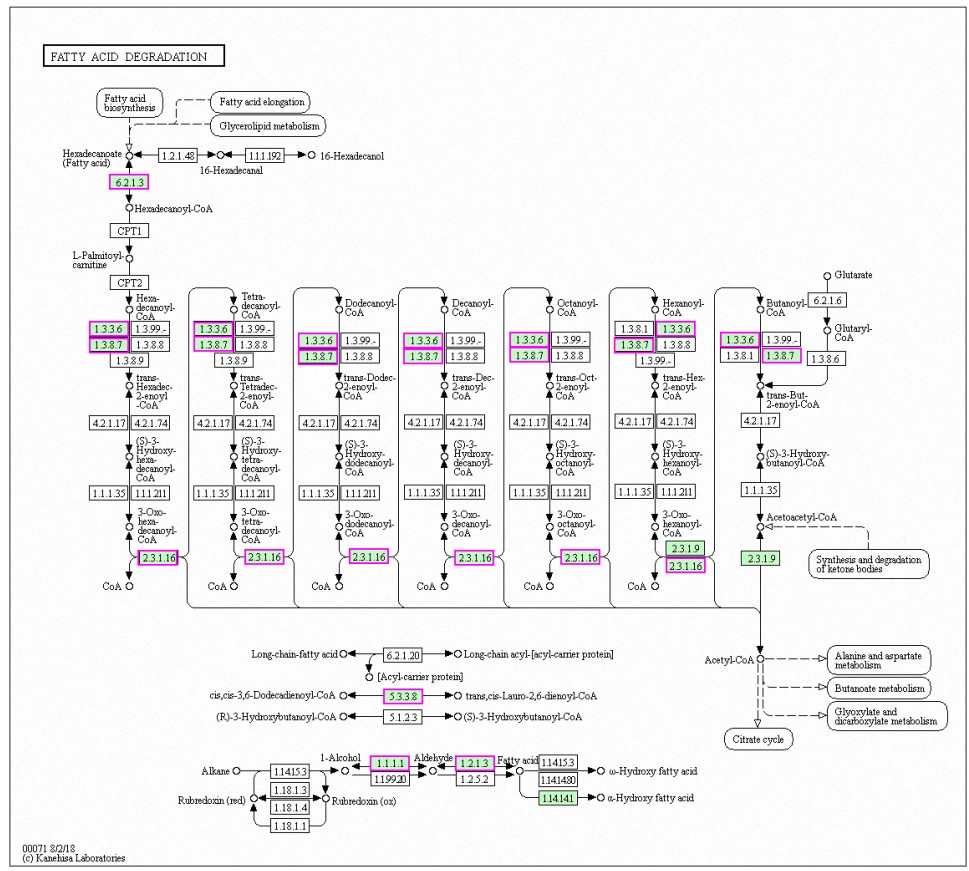

**Figure 6** **Significantly enriched KEGG pathways in fatty acid degradation.** DEGs with increased expression are marked in purple. Permission for publication was granted by the Kyoto Encyclopedia of Genes and Genomes, Kyoto University, Japan.

## Funding

This work was supported by the Science and Technology Department of Gui Zhou Province ((2019)2827, (2015)4015, (2018)5706); Doctoral Foundation of Guizhou Provincial People's Hospital (GZSYBS(2015)12; Non-profit Central Research Institute Fund of Chinese Academy of Medical Sciences (2019PT320003). The funders had no role in study design, data collection and analysis, decision to publish, or preparation of the manuscript.

## Grant Disclosures

The following grant information was disclosed by the authors:
Science and Technology Department of Gui Zhou Province: (2019)2827, (2015)4015, (2018)5706.
Doctoral Foundation of Guizhou Provincial People's Hospital: GZSYBS(2015)12.
Non-profit Central Research Institute Fund of Chinese Academy of Medical Sciences: 2019PT320003.

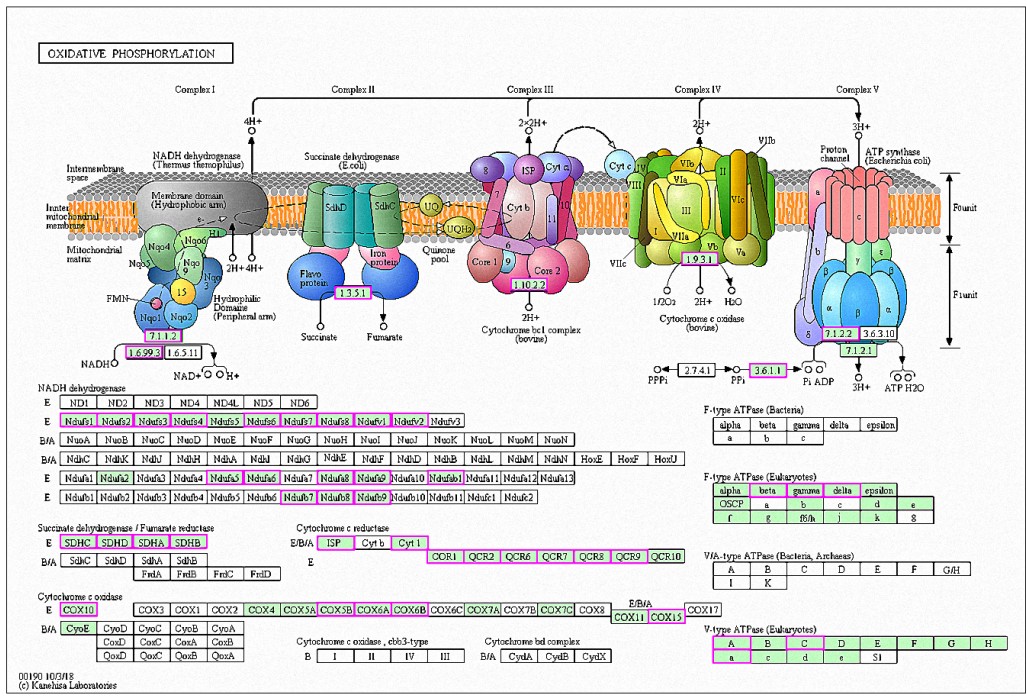

**Figure 7  Significantly enriched KEGG pathways in oxidative phosphorylation.** DEGs with increased expression are marked in purple. Permission for publication was granted by the Kyoto Encyclopedia of Genes and Genomes, Kyoto University, Japan.

## Competing Interests

The authors declare there are no competing interests.

## Author Contributions

- Rong Cheng and Zhenhua Luo conceived and designed the experiments, performed the experiments, analyzed the data, prepared figures and/or tables, authored or reviewed drafts of the paper, and approved the final draft.
- Wei Li performed the experiments, prepared figures and/or tables, and approved the final draft.
- Klarke M. Sample conceived and designed the experiments, analyzed the data, prepared figures and/or tables, and approved the final draft.
- Qiang Xu performed the experiments, authored or reviewed drafts of the paper, and approved the final draft.
- Lin Liu, Fuxun Yu and Yingjie Nie analyzed the data, authored or reviewed drafts of the paper, and approved the final draft.
- Xiangyan Zhang conceived and designed the experiments, prepared figures and/or tables, and approved the final draft.

## Data Availability

The raw data are available in the Supplementary File.

## Supplemental Information

Supplemental information for this article can be found online at http://dx.doi.org/10.7717/peerj.9767#supplemental-information.

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
