# Peer review of "Characterization of the transcriptional response of Candida parapsilosis to the antifungal peptide MAF-1A"

_PeerJ, doi:10.7717/peerj.9767_

## Round 0.1 · original submission · Major Revisions

Dear Dr Cheung - Please find reviewer comments on your article. There were a range of opinions . Reviewer 1 rejection was based on a perceived lack of novelty of using the MAF-A1 peptide against C. parapsilopsis. However, the RNA-seq approach appears to be novel and the data is unpublished. Therefore this publication is appropriate given the scope of PeerJ. The other reviewers were also more enthusiastic about the potential applications of the data.

In terms of resubmission, please consider the following guidelines

1) Consider a revised title (reviewer 2's comment)
2) Please address all specific comments raised by the reviewers
3) Consider thorough review of grammar and spelling - PeerJ will not do this on acceptance of the article
4) New experimental data is not required but can be added in response to reviewer's comments
5) Perform new data analysis where necessary.

Finally - thanks to the reviewers for returning these comments under difficult circumstances.

Reviewer 1 ·

Basic reporting

It

Experimental design

i don't think it could be explain the mechanism by RNA-seq

Validity of the findings

some problems about novelty.MAF-1A was previously reported. Antimicrobial peptides was used as antifungal is an idea that has already been explored by other investigators

Additional comments

Cheng et al. report MAF-1A report MAF-1A peptide regulates the transcriptional response ofCandida parapsilosis. While the results reported are interesting, they do not clear the novelty threshold for publication . The MAF-1A was previously reported. Antimicrobial peptides was used as antifungal is an idea that has already been explored by other investigators. i don't think the authors could demonstrate the mechanism by RNA-seq .

Reviewer 2 ·

Basic reporting

The Authors investigated the effect of the MAF-1A antifungal peptide against Candida parapsilosis. Cheng and colleagues determined the MIC value of the peptide against one of the reference strains of C. parapsilosis and determined the kinetics of growth at MIC level and compared that to the one of the peptide-free control. RNA-seq analysis was performed at two time points (6 h and 18 h) to reveal the transcriptional changes the treatment induces. They determined the differentially expressed genes and predicted protein-protein interactions. According to these changes and using KEGG pathway analysis, the Authors concluded the potential effect of MAF-1A against C. parapsilosis. The basic structure of the manuscript meets the standards of the journal.

Experimental design

Due to the emerging resistance of Candida species against the already applied antifungal agents makes this topic relevant and current, and the fundamental experimental design is correct. I found that the experiments had been carefully designed (especially the qRT-PCR analysis). Most of the methods are clear and described well.

Validity of the findings

I would recommend the Authors to be more careful when phrasing the discussion and conclusion parts of the manuscript. In summary, I think the data in this manuscript is valuable, supported with statistical analysis, but the Authors need to correct grammar mistakes, use a more elaborate English, restructure specific sections and discuss the results more carefully.

Additional comments

Due to the numerous errors in terms of grammar and style, and because of the inaccurate and incomplete presentation of the data, I do not recommend this paper to be published at this stage (see my comments later). First, I would strongly suggest the authors to perform a spellcheck by using one of the openly available online tools or to ask a native English speaker or maybe to use a dedicated service to correct these issues. There is some missing information and inconsistency in the way of presenting the literature and the results. For instance, the third paragraph of the „Introduction” mentions first AMPs, then MAF-1A and then returns to the general characterization of AMPs once again in the next paragraph. For an „Introduction” it is always advised to follow a „from the large to the specific” kind of point of view. It is also advised to mention some more AMPs from other sources. Investigation of the AMPs produced by molds really is a hot topic. To provide broader sight to the reader, it would be useful to describe how their structure and mode of action is related to the ones of AMPs from insects and specifically MAF-1A itself. It might be a convergent evolution, and they do not share much in common, but still, this would be an additional reason why to focus on AMPs produced by insects. I would have mentioned the currently applied antifungal agents, their mode of action, and how resistance have emerged against them to support better the necessity of alternative antifungal agents, such as AMPs. In the „Materials and Methods” section 2.4. subsection (Line 117) the title is „Total RNA extraction” however this paragraph also contains the description of sequencing, the title should be changed. I did not find the order of the subsections appropriate in the „Results” part either. 3.2 subsection is only 5 sentences, it could have been presented together with 3.5 subsection. The Authors might consider changing the order of the subsections as follows: 3.1. MIC assay and time-kill curves; 3.2. Transcriptional stress response and enrichment analysis of MAF-1A treated C. parapsilosis; 3.3. Verification of differentially expressed genes; 3.4. Protein-protein interaction (PPI) network analysis. I think the results part describing DEGs should be more detailed. For instance DEGs at 18 h are completely missing, and the Authors did not mention those 32 and 24 genes that were down- and upregulated at both time points respectively.
After reading the manuscript, I could not find the title appropriate. „MAF-1A peptide regulates the transcriptional repsonse of Candida parapsilosis”. The peptide regulates the response of C. parapsilosis to what? Being aware of the results I think this is an overstatement. No matter what happens to a cell, its transcriptional profile will be changed. According to the findings of Cheng et al., MAF-1A induces stress that will evidently have an effect on the global transcriptome of the fungus. Unfortunately this was not evaluated in detail in comparison with stress responses triggered in even C. albicans by other types of antifungals, that data, however, exists in the literature (See references below). Similar comparison with C. parapsilosis would be more evident, but it is obviously encumbered by the lack of such data. However the Authors mentioned in the „Introduction” part that they would also investigate the difference in the response of C. parapsilosis and C. albicans to MAF-1A, the actual comparison is also missing from the later sections of the manuscript, in turn it would strongly strengthen the quality of the paper. If the Authors have that data, I do not understand why they did not perform the analysis. These all would have provided more solid data on the MAF-1A specific fungal response and would have greatly increased the quality of the paper. Considering all these, my opinion is that this effect should not be called „regulation”. Regulation is a process by which something is specifically and tightly controlled. The described alteration in the transcriptome here seems to be a general stress response due to the interference of MAF-1A with basic biological processes (membrane integrity, protein synthesis, mitochondrial function) as the Authors concluded according to their in silico analysis. However the fungal transcriptome is likely to undergo this or similar alteration to respond these indirect effects rather than getting directly regulated due to the presence of MAF-1A. It can not be excluded however that there are genes that are MAF-1A dependently regulated (either up or down), but this can not be concluded due to the lack of the aforementioned comparisons. Regarding the title, I suggest something similar as the followings: „MAF-1A peptide alters the transcriptional profile of C. parapsilosis” or „MAF-1A peptide triggers global changes in the transcriptome of C. parapsilosis” or „Characterization of transcriptional response of C. parapsilosis to the antifungal peptide MAF-1A”. The discussion is a bit misleading, because there are AMPs mentioned from various sources that possess specific effects that were actually experimentally validated. The fact that MAF-1A treatment leads to changes in the expression of genes related to these, does not necessary mean that a 26 amino acid long peptide possesses all these properties that these peptides from different sources have.

References for transcriptome analysis:
Liu et al. (2005) Genome-Wide Expression Profiling of the Response to Azole, Polyene, Echinocandin, and Pyrimidine Antifungal Agents in Candida albicans. Antimicrobial agents and chemotherapy
Dhamgaye et al. (2012) RNA sequencing revealed novel actors of the acquisition of drug resistance in Candida albicans. BMC Genomics
And a review
Pais et al. (2019) Microevolution of the pathogenic yeasts Candida albicans and Candida glabrata during antifungal therapy and host infection. Microbial cell

Here are my remarks and comments.
It would be better to use higher resolution pictures in Figure 2; 4; 5; 6 and 7, if it is possible.
There are sentences that are difficult to understand, like:
Line 23-25 „When treated with MAF-1A from six to 18 hours, 42 genes were no longer elevated and 25 genes showed a reversed expression pattern.”
Line 52-53 „These interspecies specificities may affect host identification and clearance, in addition to antifungal drug efficacy.” Does „host identification” mean „recognition by the host”?
Line 168-169 „A total of 2439 DEGs were detected at 6 h and 18 h time points for MAF-1A treated C. parapsilosis compared to controls (Figure 2).”
Line 179-180 „The DEGs in C. parapsilosis were treated for 6 h with MAF-1A and used to construct a PPI network based on STRING.”
Line 211-212 „In recent years, C. parapsilosis infection has risen to the second or third commonly detected species of the Candida genus.”

Inconsistent spelling
Line 76, 78 RNA-seq, then Line 82 RNA seq then Line 290 and Figure 3 RNAseq
Line 102 MAF1A, the rest is MAF-1A
In the main body of the manuscript it is „qRT-PCR” but the description of Figure 3 says qPCR
Figure 3 description says Material and Methods, while in the body it is Materials and Methods
Figure 3 description still P value „=” or „<”
The title of the subsections are sometimes written as a regular sentence (only the first letter is capital) sometimes all words start with a capital.
Sometimes word with capital letters appear in the middle of the sentence e.g. Line 251 Cysteine, Line 252 Ribosomes Line 237 Ergosterol
Table S1 „Primers sequences” -> „Primer sequences”
etc.

Incorrect format or type and typos
Line 19, 36, 37, 41, 54 etc. Candida -> Candida
Line 40 C. albicans -> C. albicans
Line 18/61 Musca domestica -> Musca domestica
Line 76, 78, 81, 82, 84, 112, etc. C.albicans/parapsilosis -> C.albicans/parapsilosis
Line 77-78 An accidental „enter”
Line 88-89 „C. parapsilosis was streaked on” -> „C. parapsilosis was streaked on”
Line 258 Thermus thermophilus -> Thermus thermophilus
Line 268-269 Saccharomyces cerevisiae -> Saccharomyces cerevisiae
Line 226 C. albicans through its -> C. albicans through its
Line 135 There is a dot missing from the end of the sentence
Line 137 P valueS -> P values
Line 157 p = <0.05 Is it „equal” of „less than”?
Line 176 the Authors mention „6 h and 8 h”, it was „6 h and 18 h”
Line 194-195 „Cell cycle-yellow”, Why yellow is there?
Line 224 E. coli -> Escherichia coli
Line 254 After „coworkers” no need to write a dot.
Line 257 Bac71-35 -> Bac71-35
The right part of „Figure 4” is not visible (CPAR2_4048..)
„Conclusion” section has a different format from that of the other sections.
etc.

Space is missing
Line 113, „100µl”
Line 116 „for24h”
Line 153 „95°Cfor”
Line 231 andthat
Line 282 C.parapsilosis
Figure 2 description „….(CPBC).The expression…”
Table S1. „Table S1.Primer”
Table S2. „Table S2.The”
etc.

The first paragraph consist of five sentences (8 lines) but the word „infection” pops up eight times. I would kindly ask the Authors to rephrase the sentence, and use a more elaborate style.
References should be put at the end of the sentences. It would not break the phrase.
Version number of softwares should be put right after the name of the software without brackets, and then the reference.
Line 184-185 Supplemetary figure 1 does not refer to the enrichment analysis, is it not Supplementary Figure 6 instead?
Line 265 What is NHD?

Inappropriate phrasing
Line 213 „Some C. parapsilosis are …” -> „Certain C. parapsilosis isolates are…”
Line 215-216 „Moreover, drugs against C. parapsilosis infection have been reported in the literature.” – I do not think this sentence fits here.
Line 216 „Antimicrobial peptides have excellent antimicrobial properties” This is a self-explanatory phrasing, it should be avoided.
Figure 1/Table R1 „Time-kill curves of MAF-1A under MIC concentrations for C. parapsilosis.” It is concentration and not concentrations. There is just one MIC of a drug against one isolate.
Table 1/2 „Significantly enriched KEGG pathways for genes with increased/decreased expression after 6 h.” -> „Significantly enriched KEGG pathways for genes with increased/decreased expression after 6 h of MAF-1A treatment.”

Grammatically inaccurate phrasing (only examples)
Line 82-84 „RNA seq was used to elucidate the mechanism(s) MAF-1A and its
associated changes in gene expression during early (6 h) and late (18 h) treatment stages, investigated according to time-kill curves of C. parapsilosis growth” -> „…mechanism(s) of MAF-1A…”, „…at early (6 h) and late (18 h) time points…”
Line 164 „downwards trend for the first 8 h” -> downward trend during the first 8 h
Carefully check plural and singular form of verbs and subjects.
Figure 2. description „Volcano plots depict the log2 FC (fold change).” The dot should be a comma.
Figure 2 description still, the term ”compared to” in sentences like „The expression of 101 genes significantly increased compared to 151 genes that decreased” is misleading. Expression was compared to the control, here „in contrast to” would be more scientifically precise.
Figure 2 description still in the text it is „CPAS vs. CPAC-up”, but on the diagram it is „CPAS vs. CPAC_up”
I suggest a different colour for the venn diagram because the colour for „RG2 genes” is almost identical with the one of „CPBSvsCPBC_up” group.
Table R1 „compared with control” -> „compared to the control”
Table R1 „fungus growth e percentage” What does „e” mean?

Writing the gene names correctly
The 3.4 subsection of the „Results” summarizes the predicted protein-protein interactions, however the Authors mention „genes” in the phrases. So, are those proteins or genes? If these are proteins, then the nomenclature says, that only the first letter of the protein should be capital, the rest is minuscule like Ubi1, Cdc28 etc., and not UBI1, CDC28. On the other hand it is not suggested to use C. albicans gene/protein names for ones of C. parapsilosis. Instead, the most precise writing is as follows for genes CPAR2_809000, or CpUBI1 (and this latter need to be italicized) and for proteins „protein encoded by CPAR2_809000 (or CpUBI1)”, Candida parapsilosis Ubi1, CpUbi1 or CpUbi1p. This is also true for the „Discussion” part (e.g. Line 235 ERG1 should be CpERG1). (See nomenclature: http://www.candidagenome.org/Nomenclature.shtml#systematic)
Line 245 Improper writing of a C. parapsilosis gene. „CPAR2-807700” -> „CPAR2_807700”

Scientific questions
Why did the Authors apply 35 °C for cultivation and performing the experiments. This temperature is neither the optimum for C. parapsilosis nor it is the physiological temperature of humans.

In the „Materials and Methods” section the Authors say (Line 100) that they have adjusted the cell number to „approximately 0.5x103-2.5x103”. Why is this range? How was the cell number determined?

Line 114 „10 µl of aliquots were streaked” Was it really 10 µl and not 100? I suppose it is the latter. What does „streak” mean exactly? Was this suspension plated on an agar plate or streaked using an inoculating loop?

Line 164-165 „the curve steepened to levels comparable to the control group with 3x108 fewer colonies recorded.” Is there a calculation or comparison for the growth rates of the fungi cultivated with or without MAF-1A? What time point does „3x108 fewer colonies” refer to? These all should be presented in the results.

In the subsection 3.5 the DEGs at 18 h are not even mentioned. So were not those 24 and 32 genes that are up and down regulated at both time points respectively. How do they fit in the rest of the results?

Why did the Authors measure the absorbance at 492 nm. Absorbance measurements for Candida are always performed around 600 nm. (See references: PMID: 27627759, 29941885, 29995344)

The authors themselves have that transcriptome data of C. albicans treated with MAF-1A, why did they not include the comparison? It would strongly increase the value of the „Discussion” and „Conclusion” sections and could identify C. parapsilosis or C. albicans specific MAF-1A response.

Taking all the DEGs into account at the two time points, do these results reveal or suggest any global adaptation process to the MAF-1A treatment in C. parapsilosis?

Reviewer 3 ·

Basic reporting

This paper is easy to read and follow. English is correct.
Some revision of the figure captions are needed (Consider changing the caption of Figure 3 to explain what the figure presents. The explanation about the statistics can stay in the material and methods section). Also, does Supplementary Figure 1 relate to the protein-protein interaction or to q-PCR? I would suggest fixing the figure or the text accordingly.

Experimental design

no comment

Validity of the findings

no comment

Additional comments

In the manuscript “MAF-1A peptide regulates the transcriptional response of Candida parapsilosis”, Cheng and colleagues employ RNA-seq data to explore the mode of action of the antifungal peptide MAF-1A against C. parapsilosis. They report a number of genes to be differentially expressed after 6 and 18h of treatment. They find that genes that are differentially expressed are involved in amino acid synthesis and metabolism, oxidative phosphorylation, sterol synthesis and apoptosis. They conclude that MAF-1A’s mode of action involves disruption of membranes and normal organelle function.

This manuscript is well written and is easy to follow. Moreover, in a time when antibiotic and antifungal resistance is a widespread problem, this manuscript addresses a pressing and important issue. However, I have some comments. For instance, the authors have performed an enrichment analysis of the C.parapsilosis DEGs upon treatment with MAF-1A at the 6h timepoint and 18h timepoint. While the authors start describing the DEGs in both timepoints (section 3.2), they only describe enrichment analysis at 6h. It may be worth also performing this analysis at 18h.
It is quite interesting to note that there are some genes that are upregulated at 6h and downregulated at 18h, and all the way around. The authors focus much of their manuscript on this issue, but I feel it is equally important to discuss which genes are differentially expressed by the MAF-1A treatment itself, independently of time points (24 upregulated genes, and 32 downregulated genes, per the venn diagram).


Minor comments:

Authors chose 20 DEGs to validate RNA-Seq data by qRT-PCR. What was the rationale to choose those 20 genes?
Species names are often not italicized, both in the text and in the references

---

## Round 0.2 · Major Revisions

Although the revised manuscript answered some of the concerns, both reviewers felt that there were additional areas that had not been addressed. In particular, the request by reviewer 2 to make a comparison between the response of C. albicans and that of C. parapsilosis to MAF-1A based on publicly accessible data, was not performed. Reviewers 2 & 3 and myself agree that this is a useful study that will acceptable if appropriate revisions are made.

Reviewer 2 ·

Basic reporting

The current version of the manuscript by Rong Cheng and colleagues has been improved since the last time, however still there are some typos and sentences that are not phrased correctly. There are parts where the Authors describe the same statements in two consecutive sentences (see later). The "Introduction" section still needs some modifications. One point of the study was to propose the use of antifungal proteins (AFPs) in C. parapsilosis candidaemia. To properly strengthen this idea, it is important to give a consistent introduction on the currently available antifungal agents and more importantly the resistance emerging against them, as no such phenomenon has already been described against AFPs, that makes these molecules a potential alternative of the existing drugs in the treatment of fungaemia. I do not really recommend to mention 5-fluorocytosin as it is not used since at least two decades ago especially in monotherapy. In contrast, I really missed amphotericin B. I would also have expected to read a few sentences about how resistance emerges against these drugs that, as I mentioned before, would really support why AMPs could be alternative antifungals. I also missed a consistent introduction of AMPs including structure, chemical properties and their diverse sources from the nature. The Materials and methods section is detailed enough. The structure of the "Results" section has been modified, now it is easier to follow for the reader. Pictures are fine, however I would kindly ask the Authors to improve the resolution of Fig. 3-7, if it is possible. Regarding the qRT-PCR data I found it incomplete (Section 3.3). The Authors mentioned, that it had been performed, but no evaluation was included and this is also true for PPI network analysis (Section 3.4). I still did not find the "Discussion" and "Conclusion" sections appropriate. The Authors concluded that MAF-1A is involved in the inhibition of ergosterol synthesis, interferes with the cell membrane, protein synthesis and the function of the mitochondria. According to the analyses of GO terms and KEGG pathways it is hardly believable that one 26 amino acid long peptide is responsible for all these changes. I would recommend the Authors to try to put the results in context accordingly. I suppose some (more?) of the observed alterations are consequences of an interference with one or maybe two potential process(es), that however can not be supported without further work, therefore these statements should be phrased more carefully. Because of this, I think the last sentence of the „Discussion” part is an overstatement. Reconsider the „Discussion” and „Conclusion” parts according to the findings without statements not unequivocally supported by the results. It could be me, but I could not find a link to the raw RNAseq data, that would be essentially required to keep science transparent, hence I would kindly ask the Authors to manage this issue.

I presume that the actual target is defined by the genes whose expression remains low/high at both timepoints (UG1/2). These are oxidation-reduction processes and using alternative energy sources. This is something that could be linked to the mitochondria. In my opinion all (most) of the other changes are due to this effect. Genes upregulated at six hours show that the cell is under stress (MAPK-signaling, cell cycle, meiosis) and needs alternative energy source (via peroxisome, carbon metabolism, fatty acid degradation) and components for macromolecules (autophagy). Analysis at 18h timepont shows that the metabolism has been rewired and growth curve indicates that the growth rate of the treated suspension at the later timepoints is nearly the same as that of the untreated control, meaning that the cells could have adapted to the altered circumstances. Change in the carbon utilization processes is a well-known response of C. albicans to stress (see the following publications PMID: 17158734, 31261727, 31937647).

Experimental design

no comment

Validity of the findings

I suggested the Authors to make a comparison between the response of C. albicans and that of C. parapsilosis to MAF-1A. The Authors refused this, because "it belongs to another graduate doctor" as it is written in the rebuttal letter. I am sorry, but I can not accept this explanation, because the reference to the raw data is openly available in the paper of Wang et al. (2017) (PMID: 28567034) an article that is actually cited by the Authors (Line 84). This publication says "Sequence reads have been deposited in the NCBI Sequence Read Archive (SRA) under accession number PRJNA375109". I really urge the Authors to do this comparison, because this would serve additional valuable data on the effect of MAF-1A and moreover C. parapsilosis or C. albicans specific responses could be identified, that would be really interesting. Alternatively, rephrase the final paragraph of the introduction, because the reader will miss this comparison in the discussion, it would be so obvious.

Additional comments

I accept all the answers given to my previous questions except the one detailed at "3. Validity of findings". That analysis should either be performed or this section should be rephrased.

Grammar
Line 29 This provides new insight into the interaction between Candida parapsilosis and antimicrobial
peptides and serves as a valuable reference for future Candida parapsilosis therapies. -> This provides a new insight into the interaction between Candida parapsilosis and an antimicrobial
peptides and serves as a valuable reference for future Candida parapsilosis therapies.
Line 33-36 This sentence means the same as the following one, one of them should be deleted. And rephrase to meet proper scientific language „… (C. albicans) is the most common pathogen of Candida, its dominance has decreased as the numbers of invasive non-albicans Candida (NAC) species have increased…” -> „… (C. albicans) is the most common pathogen of Candida species, its dominance has decreased as the incidence of non-albicans Candida (NAC) species have increased…”
Line 41 their should be its (it refers to C. prapsilosis, it is singular)
Line 47 Rates of infection are high -> Rate of infection is high
Line 48-49 … and drug resistance conrast those of … -> … and drug resistance are in conrast with those of …
Line 53 „… immune response for a variety …” -> „… immune response of a variety …”
Line 54-56 This sentence is grammatically incorrect, please check it.
Line 108 triplicate -> triplicates
Line 159 A dot is missing before the word „Primers”
Line 269 „… transport process on the mitochondrial …” -> „… transport process in the mitochondrial …”
Line 281 „These processes help cells remove accumulated …” ->„These processes help cells to remove accumulated …”


Typos
Check for double spaces and missing spaces.
Line 53 The Authors write „from” is it not „form” ?
Maybe it is only the formatting of the homepage of the journal, but neither of the species names and the latin words are in italic.
Line 225. A dot is missing from the end of the senctence.

Inconsistent phrasing
Line 21-22 Instead of „DEGs” use „genes”. An upregulated/downregulated gene is a differentially expressed gene.
Line 45-46 Recent epidemiological studies (in various geographical regions) have shown that C. parapsilosis is the second to third most prevalent infection, following only C. albicans (Toth et al. 2019). -> The statement is correct if C. parapsilosis is the seconds, then C. albicans is the first. But if C. parapsilosis is the third then there must be some other species besides C. albicans. Please rephrase this sentence.
Comment on line 49: It should not be „may”, the interspecies specificities do actually affect the recognition be the host (C. parapsilosis vs C. albicans, see reference PMID: 26793173)
Line 64 Eefflux pumps are drug resistance related genes, if the latter is written, no need to mention the other.
In text UG1 was mentioned as significant (Line 201), however it is missing from Table 6.
Line 243-245 If overexpression is mentioned, point mutation should be as well (PMID: 23480635, 25385095).
Line 274-276 This sentence is misleading. The goal of ROS production is not the disruption of the electron transport chain, but it is a side effect of their presence.
Line 276 If the term „ROS” is mentioned, then no need to write hydrogen peroxide and hydroxy radicals, because these are defined as ROS.

Style
Line 51 Common Candida pathogens -> Remove „Common”
Line 54 A scientific paper should not contain such expression as „relatively small”.
Line 64 Instead of „resistant gene” I would recommend to use „drug resistance related gene”
Line 115 Time-kill curves were adapted from Li et al and Sun et al -> Instead of mentioning the complete citation in the sentence, the Authors might want to consider writing „Time-kill curves were performed according to the literature” instead, and then the citation can be included.
Citation should be at the very end of a sentence (e.g. Line 140 and 141)
Line 163-165 This sentence means the same as the following two. This part should be modified.
Line 166-167 It is not the curve that increased or remained, but the cell number. This should be rephrased properly.
Line 169 „RNA-seq results from C. parapsilosis with MAF-1A after …” -> „RNA-seq results from C. parapsilosis treated with MAF-1A after …”
Line 170 Instead of „Of the genes …” I recommend to write „Out of these genes …” or „Out of the DEGs …”
Line 171 Instead of „… 4.38% of the genes.” I suggest to write „… 4.38% of the total expressed genes.”
Line 174-176 Repharase sentence to make it grammatically correct, it has two subjects.
Line 218 „second or third commonly” -> „second or third most commonly”
Line 261 „… on ribosomes, inhibiting translation” -> „… on ribosomes therefore inhibiting translation”
Line 264 „… resulted in a loss of expression of multiple genes involved with ribosomes and translational processes.” -> „… resulted in a loss of expression of multiple genes associated with ribosomes and translational processes.”
Figure 2 „The expression of 101 genes significantly increased in contrast to 151 genes that decreased (padj < 0.05).” It is the expression that decreased not the genes. „The expression of 101 genes significantly increased in contrast to 151 genes whose expression decreased (padj < 0.05).”
Figure 2 „Gene expression venn diagrams revealing two gene groups with opposite trends, labeled as RG1, RG2, and two gene groups with independently of time points, labeled as UG1 UG2; CPAS vs. CPAC_up: genes with increased expression after 6 hours; CPAS vs. CPAC_down: genes with decreased expression after 6 hours; CPBS vs. CPBC_up: genes with increased expression after 18 hours; CPBS vs. CPBC_down: genes with decreased expression after 18 hours.” Please repharase this sentence, maybe it should be split into more senteces. The tense is incorrect, if it is present continuous „is” is missing, but actually it should be simple present: „reveals”. The expression „two gene groups with independently of time points” does not have a meaning.

It is inconsistent that Fig 5 shows enzymes by highlighting the whole box where their names are, while on Fig 6 and 7 a purple frame is used to sign the proteins encoded by genes whose expression have changed. Could it be standardized?

Reviewer 3 ·

Basic reporting

no comment

Experimental design

no comment

Validity of the findings

no comment

Additional comments

no comment

---

## Round 0.3 · accepted · Accept

Please take care of the minor editing tasks required by the reviewer.

Reviewer 3 ·

Basic reporting

no comment

Experimental design

no comment

Validity of the findings

no comment

Additional comments

I think the authors have addressed the main comments expressed by the reviewers. I noticed, however, that while gene names are italicized, species names remain unchanged both in the text and in the references.